# Impact of gender on mid-term prognosis of patients undergoing coronary artery bypass grafting

**Woo Jin Jang**[1‡], **Ki Hong Choi**[2‡], **Jihoon Kim**[2], **Jeong Hoon Yang**[2], **Joo-Yong Hahn**[2], **Seung-Hyuk Choi**[2], **Hyeon-Cheol Gwon**[2], **Yang Hyun Cho**[3], **Kiick Sung**[3], **Wook Sung Kim**[3], **Dong Seop Jeong**[3], **Young Bin Song**[2]*

1 Division of Cardiology, Department of Internal Medicine, Seoul Hospital, Ewha Womans University College of Medicine, Seoul, Republic of Korea, 2 Division of Cardiology, Department of Medicine, Heart Vascular Stroke Institute, Samsung Medical Center, Sungkyunkwan University School of Medicine, Seoul, Republic of Korea, 3 Department of Thoracic and Cardiovascular Surgery, Heart Vascular Stroke Institute, Samsung Medical Center, Sungkyunkwan University School of Medicine, Seoul, Republic of Korea

‡ WJJ and KHC contributed equally as first authors to this work.
* youngbin.song@gmail.com

**Data Availability Statement:** All relevant data are within the manuscript and its Supporting information files.

## Abstract

### Objectives

We evaluated the impact of sex on mid-term prognosis in patients who underwent coronary artery bypass grafting (CABG). Data on gender differences in current management or clinical outcomes after CABG are controversial, and there have been limited data focusing on them.

### Methods

This was a retrospective and prospective, single-center, observational study. Between January 2001 and December 2017, 6613 patients who underwent CABG were enrolled from an institutional registry of Samsung Medical Center, Seoul, Korea (Clinicaltrials.gov, NCT03870815) and divided into two groups according to sex (female group, n = 1679 vs. male group, n = 4934). The primary outcome was cardiovascular death or myocardial infarction (MI) at 5 years. Propensity score matching analysis was performed to reduce confounding factors.

### Results

During a mean follow-up duration of 54 months, a total of 252 cardiovascular death or MIs occurred (female, 78 [7.5%] vs. male, 174 [5.7%]). Multivariate analysis revealed no significant difference in the incidence of cardiovascular death or MI at 5 years between female and male groups (hazard ratio [HR] 1.05; 95% confidence interval [CI] 0.78 to 1.41; $p$ = 0.735). After propensity score matching, the incidence of cardiovascular death or MI was still similar between the two groups (HR 1.08; 95% CI 0.76 to 1.54; $p$ = 0.666). The similarity of long-term outcomes between the two groups was consistent across various subgroups. There was also no significant difference in the risk of 5-year cardiovascular death or

**Funding:** The authors received no specific funding for this work.

**Competing interests:** The authors have declared that no competing interests exist.

**Abbreviations:** BARC, bleeding academic research consortium; **CABG**, coronary artery bypass grafting; **CI**, confidence interval; **HR**, hazard ratio; **MACE**, major adverse cardiac event; **MI**, myocardial infarction; **PCI**, percutaneous coronary intervention.

MI between males and females according to age (pre- and postmenopausal status) (*p* for interaction = 0.437).

## Conclusions

After adjusting for baseline differences, sex does not appear to influence long-term risk of cardiovascular death or MI in patients undergoing CABG.

## Clinical trials.gov number

NCT03870815.

## Introduction

Coronary artery bypass grafting (CABG) is an essential treatment in patients with complex or multi-vessel coronary artery disease [1]. Differences exist in presentation patterns, lesion characteristics, clinical outcomes, and response to therapy between male and female patients with cardiovascular disease [2]. However, data on sex differences in clinical outcomes after CABG are controversial, and there have been limited data focusing on long-term outcomes. Several studies reported that women suffering from coronary heart disease tend to have more co-morbidities and are more likely to present with a severe form of cardiogenic shock that requires urgent or emergent CABG compared with men [3–5]. Moreover, women have relatively small body surface area, and thus smaller coronary vessels. This leads to technical difficulties during CABG and makes female patients prone to higher surgical risk and poorer post-surgical outcome than male patients [6–9]. On the other hand, other studies have reported no differences in clinical outcomes based on sex after CABG [10–12]. In particular, considering the improvements in CABG outcomes in recent decades due to advanced surgical techniques, improved post-operative care, and new secondary prevention medications [13–15], there is a need for re-evaluation of the long-term prognostic role of sex in CABG outcomes. Therefore, we assessed the impact of sex on mid-term clinical outcomes in patients who underwent CABG using a large, dedicated, and recent real-world registry.

## Methods

### Study population and data collection

This study was a retrospective and prospective, single-center, observational registry of patients with coronary artery disease who underwent CABG (Clinicaltrials.gov, NCT03870815). The study design and population have been described in detail previously [16]. In brief, between January 2001 and December 2017, a total of 6,691 consecutive patients were enrolled from Samsung Medical Center, Seoul, and Republic of Korea. For this study, 14 patients who were younger than 18 years of age and 64 patients who did not have available angiography or surgery data were excluded from analysis. Data from the remaining 6,613 patients were evaluated, and the subjects were divided into two groups according to sex (Fig 1).

Baseline characteristics, angiographic data, surgical procedural data, and outcome data were collected prospectively from the registry by research coordinators. Additional information was obtained from medical records and telephone interviews, if necessary. Mortality data for patients who were lost to follow-up were confirmed by National Death Records. All events were adjudicated by a cardiology expert with blinded fashion. The study protocol was

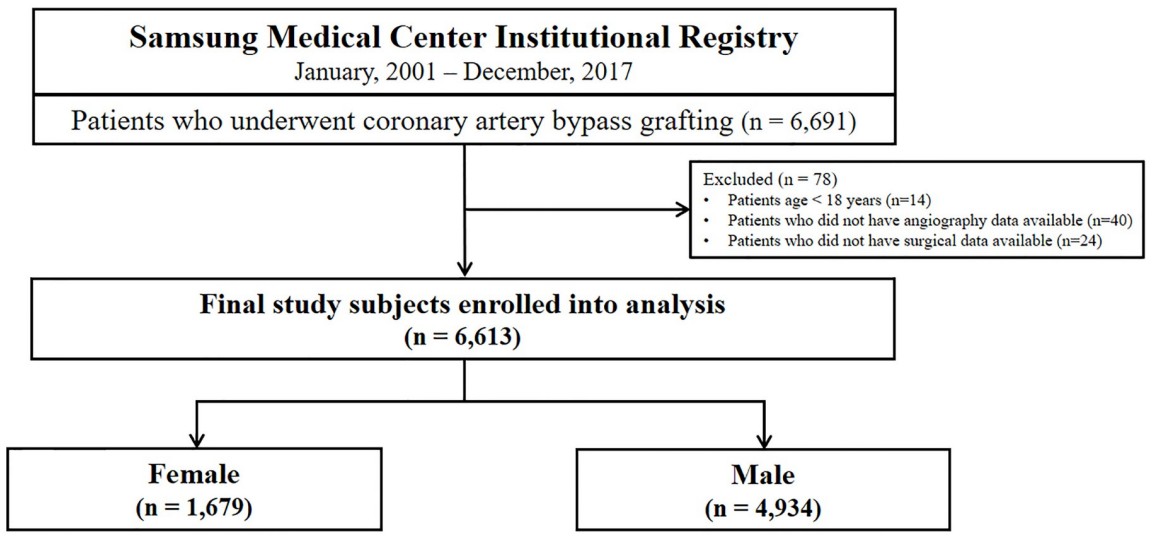

**Fig 1. Study flow chart.**

approved and the requirement for informed consent from the individual patients was waived by the Institutional Review Board of Samsung Medical Center. This study was conducted according to the principles of the Declaration of Helsinki.

## Surgical techniques

CABG was performed in accordance with relevant standard guidelines [1]. All operations were performed through standard median sternotomy. Bilateral internal thoracic arteries were prepared by skeletonization techniques with sharp dissection, clipping, and branch ligation. The saphenous vein was harvested from the patient's upper or lower leg via split incisions. Heparinized saline was used to dilate the diameter of the saphenous vein graft (SVG). The in situ right gastroepiploic artery was prepared in a pedicled manner. The right internal thoracic artery (RITA) was anastomosed to the left side of the left internal thoracic artery (LITA) with a continuous running suture to construct a Y composite graft. With few exceptions, the LITA was anastomosed to the left anterior descending artery and its branches, and the RITA was sequentially anastomosed to the left circumflex artery. The RITA was initially selected as a graft if proximal right coronary artery stenosis was greater than 80%. If the length of the harvested RITA was not sufficient to reach the right coronary artery anastomosis, the right gastroepiploic artery was used. If the proximal right coronary artery stenosis was less than 80%, aortocoronary bypass was performed using the SVG. A Transonic Flowmeter (Transonic Systems, Ithaca, NY, USA) was used to evaluate the quality of the anastomosis according to the transit time flow rate. Off-pump CABG utilizing the bilateral internal thoracic artery is the preferred technique at our institution. Perioperative treatment strategies including the use of cardiopulmonary bypass, number of grafts used, determination of anastomosis site, and use of concomitant medications after CABG were all at the discretion of individual operators.

## Study outcomes

The primary outcome of the present study was the occurrence of cardiovascular death or myocardial infarction (MI) at 5 years. Secondary outcomes included cardiovascular death; all-

cause death; MI; stroke; Bleeding Academic Research Consortium (BARC) type 3–5 bleeding; and a major adverse cardiovascular event (MACE) defined as a composite of cardiovascular death, MI, or stroke. All deaths were considered to be of cardiac origin unless a definite non-cardiac cause could be established. MI was defined as an elevated cardiac troponin or myocardial band fraction of creatine kinase greater than the upper reference limit with concomitant ischemic symptoms or electrocardiography findings indicative of ischemia. Procedure-related MI was not included in this definition of MI. Stroke was defined as any non-convulsive focal or global neurological deficit of abrupt onset caused by ischemia or hemorrhage within the brain.

## Statistical analysis

Continuous variables were compared using the Student's $t$ test or the Wilcoxon rank-sum test where applicable and are presented as mean ± standard deviation or as the median with interquartile range. Categorical data were compared between groups using the Fisher's exact test or the chi-square test, as appropriate, and are presented as numbers and relative frequencies (%). Cumulative event rates were estimated using the Kaplan-Meier method, and treatment effects were assessed by stratified log-rank statistics. Patients were censored at 5 years (1,825 days) or when events occurred. After stratifying patients who underwent CABG by sex, clinical outcomes were compared between female and male groups using a Cox proportional hazard regression model to calculate hazard ratios (HRs) and 95% confidence intervals (CIs). Proportional hazards assumptions of the HR for females compared with males in the Cox proportional hazards models were graphically inspected in the "log minus log" plot and were also confirmed with the Schoenfeld residual test. Adjusted HRs and 95% CIs for clinical outcomes according to sex were obtained by the final Cox regression model that included age, body mass index, hypertension, diabetes mellitus, current smoking, heart failure, previous history of MI, use of antiplatelet medication, use of beta-blockers, use of angiotensin converting enzyme inhibitor or angiotensin receptor blockers, use of statins, multi-vessel disease, left main involvement, off-pump CABG, combined valvular surgery, number of anastomoses, use of LITA, use of RITA, use of bilateral internal thoracic arteries, and use of SVG. Propensity-score matched analysis was also performed to reduce selection bias. The covariate balance after propensity-score matching was assessed by calculating absolute standardized mean differences. Standardized mean differences after propensity-score matching were within ± 10% across all matched covariates with variance ratios near 1.0, suggesting achievement of balance between the female and male groups. Stratified Cox proportional hazard models were used to compare the outcomes of the matched groups. All probability values were two-sided, and $p < 0.05$ was considered to be statistically significant. Statistical analyses were performed using R Statistical Software (version 3.6.0; R Foundation for Statistical Computing, Vienna, Austria).

## Results

### Baseline characteristics

A total of 6,613 patients that were treated with CABG were recruited for inclusion in the current study. The mean age and body mass index of the study population were 63.5 ± 9.8 years and 24.6 ± 3.0, respectively. Among the overall population, 1,679 patients were female (25.4%, female group) and 4,934 were male (74.6%, male group). Female patients were older, had lower body mass index, and reported less smoking than male patients (Table 1). The female group had a significantly higher incidence of hypertension and diabetes mellitus, but a lower incidence of peripheral vascular disease and heart failure compared with the male group. In addition, female patients were more likely to present with acute coronary syndrome (61.8% of

**Table 1. Baseline clinical characteristics.**

| | Overall population (n = 6,613) | | | | Propensity-matched population (n = 3,108, 1,554 pairs) | | | |
|---|---|---|---|---|---|---|---|---|
| | **Female** | **Male** | *P* value | SMD | **Female** | **Male** | *P* value | SMD |
| | **n = 1,679** | **n = 4,934** | | | **n = 1,554** | **n = 1,554** | | |
| Age *(years)* | 66.9 ± 8.6 | 62.3 ± 9.9 | <0.001 | -0.460 | 66.4 ± 8.5 | 66.1 ± 9.0 | 0.408 | -0.026 |
| Body mass index *(kg/m²)* | 24.3 ± 3.3 | 24.7 ± 2.9 | <0.001 | 0.121 | 24.4 ± 3.3 | 24.4 ± 2.8 | 0.949 | -0.002 |
| Hypertension | 1,199 (71.4) | ,2959 (60.0) | <0.001 | -0.234 | 1,084 (69.8) | 1,068 (68.7) | 0.560 | -0.021 |
| Diabetes mellitus | 851 (50.7) | 2,147 (43.5) | <0.001 | -0.145 | 764 (49.2) | 763 (49.1) | 1.000 | -0.001 |
| Chronic kidney disease | 109 (6.5) | 338 (6.9) | 0.653 | 0.014 | 103 (6.6) | 107 (6.9) | 0.830 | 0.011 |
| Dyslipidemia | 507 (30.2) | 1,441 (29.2) | 0.460 | -0.022 | 450 (29.0) | 471 (30.3) | 0.432 | 0.029 |
| Peripheral vascular disease | 75 (4.5) | 394 (8.0) | <0.001 | 0.129 | 75 (4.8) | 84 (5.4) | 0.515 | 0.021 |
| Heart failure | 439 (26.1) | 1,526 (30.9) | <0.001 | 0.103 | 412 (26.5) | 406 (26.1) | 0.839 | -0.008 |
| LVEF, before CABG *(%)* | 56.8 ± 13.4 | 54.0 ± 13.7 | <0.001 | -0.201 | 56.4 ± 13.4 | 56.4 ± 12.9 | 0.891 | -0.004 |
| Current smoking | 101 (6.0) | 2,052 (41.6) | <0.001 | 0.722 | 101 (6.5) | 112 (7.2) | 0.478 | 0.014 |
| Previous history of PCI | 325 (19.4) | 957 (19.4) | 0.972 | 0.001 | 306 (19.7) | 326 (21.0) | 0.397 | 0.032 |
| Previous history of MI | 158 (9.4) | 592 (12.0) | 0.004 | 0.079 | 146 (9.4) | 153 (9.8) | 0.715 | 0.013 |
| Previous history of CABG | 20 (1.2) | 67 (1.4) | 0.694 | 0.014 | 19 (1.2) | 18 (1.2) | 1.000 | -0.005 |
| Previous history of stroke | 234 (13.9) | 708 (14.3) | 0.706 | 0.012 | 217 (14.0) | 233 (15.0) | 0.444 | 0.029 |
| *Clinical presentation* | | | | | | | | |
| Stable ischemic heart disease | 641 (38.2) | 2320 (47.0) | <0.001 | 0.177 | 629 (40.5) | 651 (41.9) | 0.691 | 0.028 |
| Unstable angina | 710 (42.3) | 1687 (34.2) | | | 635 (40.9) | 626 (40.3) | | |
| Acute myocardial infarction | 328 (19.5) | 927 (18.8) | | | 290 (18.7) | 277 (17.8) | | |
| *Concomitant medical treatment* | | | | | | | | |
| Antiplatelets *(Aspirin or [1]P2Y12 inhibitor)* | 1,660 (98.9) | 4,903 (99.4) | 0.058 | 0.063 | 1,538 (99.0) | 1,538 (99.0) | 1.000 | 0.001 |
| Beta-blockers | 1,158 (69.0) | 3,573 (72.4) | 0.008 | 0.077 | 1,084 (69.8) | 1,067 (68.7) | 0.534 | -0.024 |
| ACE inhibitors or ARBs | 680 (40.5) | 1,617 (32.8) | <0.001 | -0.164 | 598 (38.5) | 579 (37.3) | 0.506 | -0.026 |
| Statins | 1,241 (73.9) | 3,870 (78.4) | <0.001 | 0.109 | 1,161 (74.7) | 1,169 (75.2) | 0.772 | 0.012 |

Data are presented as mean ± standard deviation or n (%).

[1]P2Y12 inhibitors included clopidogrel, ticagrelor, and prasugrel.

ACE = angiotensin converting enzyme, ARB = angiotensin receptor blocker, CABG = coronary artery bypass grafting surgery, LVEF = left ventricular ejection fraction, MI = myocardial infarction, PCI = percutaneous coronary intervention, SMD = standardized mean difference

the female group) and had a higher frequency of previous MI compared with male patients. Antiplatelet medications, beta-blockers, and statins were prescribed less frequently in the female group, but use of angiotensin converting enzyme inhibitors or angiotensin receptor blockers was more frequent (Table 1).

Coronary anatomy and procedural characteristics are shown in Table 2. The proportion of urgent or emergent CABG was not significantly different between the female and male groups. The proportion of off-pump CABG performance was significantly lower in the female group than in the male group, but the proportion of combined valvular surgery was higher in the female group compared with the male group. The incidence of multi-vessel disease and left main disease was lower in the female group than in the male group. The use of arterial grafts including LITA, RITA, and bilateral internal thoracic arteries was less frequent but the use of SVG was more frequent in the female group than in the male group (Table 2).

## Clinical outcomes

**Overall population.** In-hospital mortality was significantly higher in the female group than in the male group (2.0% in the female group vs. 1.0% in the male group, *p* value = 0.002),

Table 2. Coronary anatomy and procedural characteristics.

| | Overall population (n = 6,613) | | | | Propensity-matched population (n = 3,108, 1,554 pairs) | | | |
|---|---|---|---|---|---|---|---|---|
| | **Female** | **Male** | *P* value | SMD | **Female** | **Male** | *P* value | SMD |
| | **n = 1,679** | **n = 4,934** | | | **n = 1,554** | **n = 1,554** | | |
| Urgent/Emergent CABG | 152 (9.1) | 379 (7.7) | 0.083 | -0.051 | 132 (8.5) | 130 (8.4) | 0.949 | -0.004 |
| Multi-vessel disease | 1,533 (91.3) | 4,626 (93.8) | 0.001 | 0.101 | 1,428 (91.9) | 1,429 (92.0) | 1.000 | 0.002 |
| Left main involvement | 289 (17.2) | 1037 (21.0) | 0.001 | 0.093 | 274 (17.6) | 268 (17.2) | 0.813 | -0.009 |
| *Number of diseased coronary vessel* | | | | | | | | |
| Three-vessel disease | 1,154 (68.7) | 3,433 (69.6) | 0.535 | 0.018 | 1,071 (68.9) | 1,059 (68.1) | 0.671 | -0.016 |
| Two-vessel disease | 379 (22.6) | 1,193 (24.2) | 0.193 | 0.037 | 357 (23.0) | 370 (23.8) | 0.611 | 0.019 |
| Single-vessel disease | 146 (8.7) | 308 (6.2) | 0.001 | -0.101 | 126 (8.1) | 125 (8.0) | 1.000 | -0.002 |
| Off-pump CABG | 1,336 (79.6) | 4,040 (81.9) | 0.039 | 0.061 | 1,254 (80.7) | 1,272 (81.9) | 0.434 | 0.029 |
| Combined valvular surgery | 135 (8.0) | 289 (5.9) | 0.002 | -0.093 | 113 (7.3) | 121 (7.8) | 0.634 | 0.021 |
| Number of anastomosis sites | 3.7 ± 1.3 | 3.9 ± 1.3 | <0.001 | 0.178 | 3.7 ± 1.3 | 3.7 ± 1.3 | 0.775 | -0.011 |
| *Used graft vessel* | | | | | | | | |
| Left internal thoracic artery | 1,602 (95.4) | 4,784 (97.0) | 0.003 | 0.091 | 1,494 (96.1) | 1,490 (95.9) | 0.783 | -0.015 |
| Right internal thoracic artery | 1,352 (80.5) | 4,236 (85.9) | <0.001 | 0.152 | 1,272 (81.9) | 1,273 (81.9) | 1.000 | 0.001 |
| Bilateral thoracic arteries | 1,321 (78.7) | 4,162 (84.4) | <0.001 | 0.156 | 1,246 (80.2) | 1,248 (80.3) | 0.964 | 0.003 |
| [1]Other arterial grafts | 246 (14.7) | 728 (14.8) | 0.950 | 0.002 | 228 (14.7) | 214 (13.8) | 0.504 | -0.025 |
| Saphenous vein graft | 398 (23.7) | 928 (18.8) | <0.001 | -0.125 | 347 (22.3) | 332 (21.4) | 0.543 | -0.024 |

Data are presented as mean ± standard deviation or n (%).

[1]Other arterial grafts included radial artery and right gastroepiploic artery.

CABG = coronary artery bypass grafting surgery, SMD = standardized mean difference

as well as 30-day mortality (1.4% vs. 0.8%, *p* value = 0.025). During a mean follow-up of 54 months, a total of 252 cardiovascular death or MIs occurred. In multivariate analysis, there was no significant difference in the incidence of cardiovascular death or MI at 5 years (7.5% in the female group vs. 5.7% in the male group, adjusted HR 1.05, 95% CI 0.78–1.41; *p* value = 0.735) between the female and male groups. The incidence of cardiovascular death (6.1% vs. 4.4%, adjusted HR 1.06, 95% CI 0.77–1.47; *p* value = 0.710), all-cause death (10.9% vs. 8.7%, adjusted HR 0.95, 95% CI 0.75–1.20; *p* value = 0.661), MI (1.8% vs. 1.7%, adjusted HR 1.01, 95% CI 0.54–1.91; *p* value = 0.970), stroke (4.2% vs. 4.0%, adjusted HR 0.96, 95% CI 0.66–1.40; *p* value = 0.825), BARC type 3–5 bleeding (3.6% vs. 3.2%, adjusted HR 0.97, 95% CI 0.68–1.39; *p* value = 0.867), and MACE (10.9% vs. 9.2%, adjusted HR 0.99, 95% CI 0.78–1.25; *p* value = 0.927) was similar between the two groups (Fig 2 and Table 3).

**Propensity-matched population.** After performing propensity score matching, a total of 1,554 pairs were generated, and there were no significant differences in baseline clinical or angiographic characteristics for the propensity score-matched subjects (Tables 1 and 2). A total of 198 cardiovascular death or MIs occurred during follow-up in matched patients, and there was no significant difference in the incidence of cardiovascular death or MI at 5 years (matched HR 1.08, 95% CI 0.76–1.54; *p* value = 0.666) between the female and male groups. The risk of cardiovascular death (matched HR 1.14, 95% CI 0.78–1.67; *p* value = 0.507), all-cause death (matched HR 1.02, 95% CI 0.77–1.34; *p* value = 0.904), MI (matched HR 0.94, 95% CI 0.43–2.05; *p* value = 0.868), stroke (matched HR 0.99, 95% CI 0.64–1.52; *p* value = 0.947), BARC type 3–5 bleeding (matched HR 0.98, 95% CI 0.65–1.48; *p* value = 0.927), and MACE (matched HR 1.03, 95% CI 0.78–1.36; *p* value = 0.839) were also similar between the two groups (Fig 2 and Table 3).

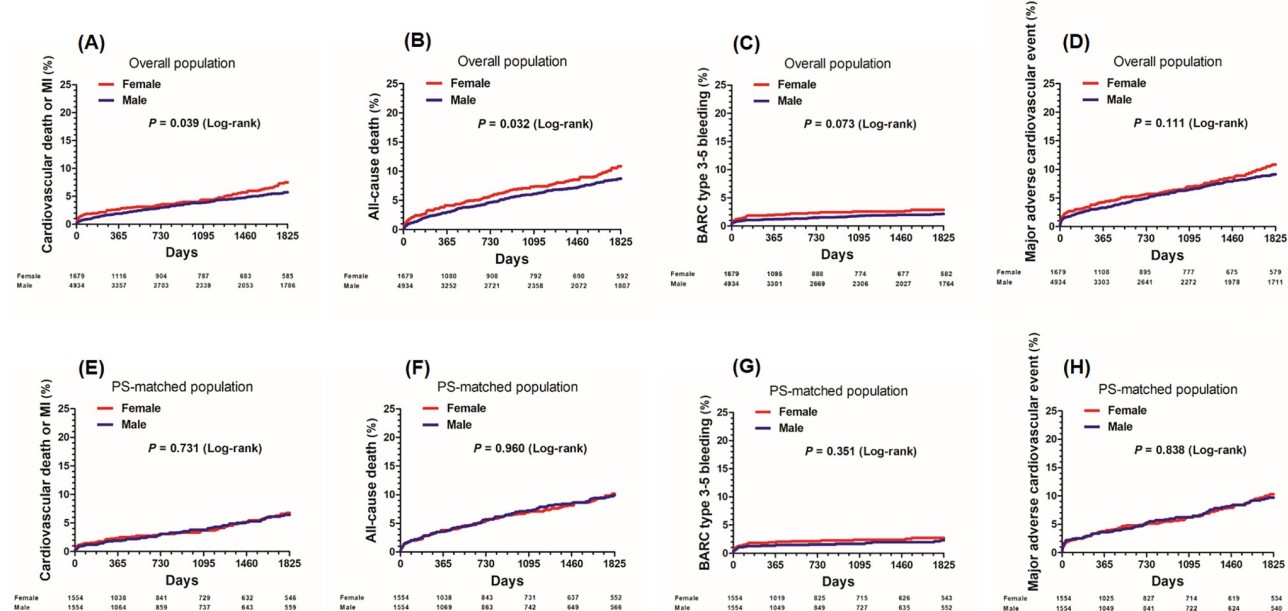

**Fig 2. Kaplan-Meier curves comparing 5-year risk of clinical outcomes according to sex.** CABG = coronary artery bypass grafting, BARC = bleeding academic research consortium, MI = myocardial infarction, PS = propensity score.

### Sex disparities according to menopausal age

Among the study population, there were 1,344 patients under 55 years of age (18 to 55 years old; premenopausal age) and 5,269 patients over 56 years (postmenopausal age). In patients under 55 years of age, there were no significant differences in the incidence of cardiovascular death or MI at 5 years (1.2% in female vs. 2.0% in male, adjusted HR 0.56, 95% CI 0.12–2.51; $p$ value = 0.447) between the female and male subgroups. In patients over 56 years of age, the rate of cardiovascular death was similar between the two sex subgroups (5.0% vs. 4.0%, adjusted HR 1.08, 95% CI 0.79–1.46; $p$ value = 0.645). In the female subgroup, post-meno-pausal-aged patients had a higher incidence of cardiovascular death or MI at 5 years com-pared to pre-menopausal-aged patients (1.2% vs. 5.0% adjusted HR 0.22, 95% CI 0.05–0.89; $p$ value = 0.034) (Fig 3).

### Subgroup analysis

To investigate the association between sex differences and cardiovascular death or MI after CABG in various subgroups, we performed subgroup analyses. Sex did not differ significantly across subgroups regardless of clinical presentation (acute coronary syndrome vs. stable ische-mic heart disease), comorbidities (diabetes vs. non-diabetes and left ventricular dysfunction or not), surgical characteristics (urgent or emergent CABG vs. elective CABG and off-pump CABG vs. on-pump CABG), lesion location (presence or absence of left main involvement), or used graft vessel (arterial graft only vs. arterial graft and SVG) (Fig 4).

### Discussion

We investigated the effect of sex differences on mid-term clinical outcomes in patients who underwent CABG using a large, dedicated, recent real-world CABG registry. Our main finding was that there was no significant difference in mid-term risk of cardiovascular death or MI at 5

**Table 3. Comparison of 5-year risk of clinical outcomes.**

|  | Female | Male | Univariable analysis | | | [1]Multivariable analysis | | | Propensity-matched analysis | | |
|---|---|---|---|---|---|---|---|---|---|---|---|
|  | | | HR | 95% CI | P value | HR | 95% CI | P value | HR | 95% CI | P value |
|  | n = 1,679 | n = 4,934 | | | | | | | n = 3,108 (1,554 pairs) | | |
| Cardiovascular death or myocardial infarction | 78 (7.5) | 174 (5.7) | 1.34 | 1.03–1.75 | 0.031 | 1.05 | 0.78–1.41 | 0.735 | 1.08 | 0.76–1.54 | 0.666 |
| Cardiovascular death | 66 (6.1) | 136 (4.4) | 1.45 | 1.08–1.95 | 0.013 | 1.06 | 0.77–1.47 | 0.710 | 1.14 | 0.78–1.67 | 0.507 |
| All-cause death | 116 (10.9) | 272 (8.7) | 1.28 | 1.03–1.59 | 0.026 | 0.95 | 0.75–1.20 | 0.661 | 1.02 | 0.77–1.34 | 0.904 |
| Myocardial infarction | 16 (1.8) | 46 (1.7) | 1.04 | 0.59–1.84 | 0.882 | 1.01 | 0.54–1.91 | 0.970 | 0.94 | 0.43–2.05 | 0.868 |
| Stroke | 44 (4.2) | 129 (4.0) | 1.02 | 0.72–1.43 | 0.930 | 0.96 | 0.66–1.40 | 0.825 | 0.99 | 0.64–1.52 | 0.947 |
| BARC type 3–5 bleeding | 50 (3.6) | 130 (3.2) | 1.14 | 0.82–1.58 | 0.438 | 0.97 | 0.68–1.39 | 0.867 | 0.98 | 0.65–1.48 | 0.927 |
| [2]MACE | 116 (10.9) | 291 (9.2) | 1.19 | 0.96–1.47 | 0.117 | 0.99 | 0.78–1.25 | 0.927 | 1.03 | 0.78–1.36 | 0.839 |

Values are n (%). Cumulative incidence of events was presented as Kaplan–Meier estimates.

ACE = angiotensin converting enzyme, ARB = angiotensin receptor blocker, BARC = bleeding academic research consortium, CABG = coronary artery bypass grafting surgery, CI = confidence interval, HR = hazard ratio, LITA = left internal thoracic artery, MACE = major adverse cardiovascular event, MI = myocardial infarction, RITA = right internal thoracic artery, SVG = saphenous vein graft.

[1] Adjusted variables included age, body mass index, hypertension, diabetes mellitus, current smoking, heart failure, previous history of MI, use of antiplatelet, use of beta-blocker, use of ACE inhibitor or ARB, use of statin, multi-vessel disease, left main involvement, off-pump CABG, combined valvular surgery, number of anastomosis, use of LITA, use of RITA, use of bilateral thoracic arteries, and use of SVG.

[2] MACE was defined as the composite of cardiovascular death, myocardial infarction, and stroke.

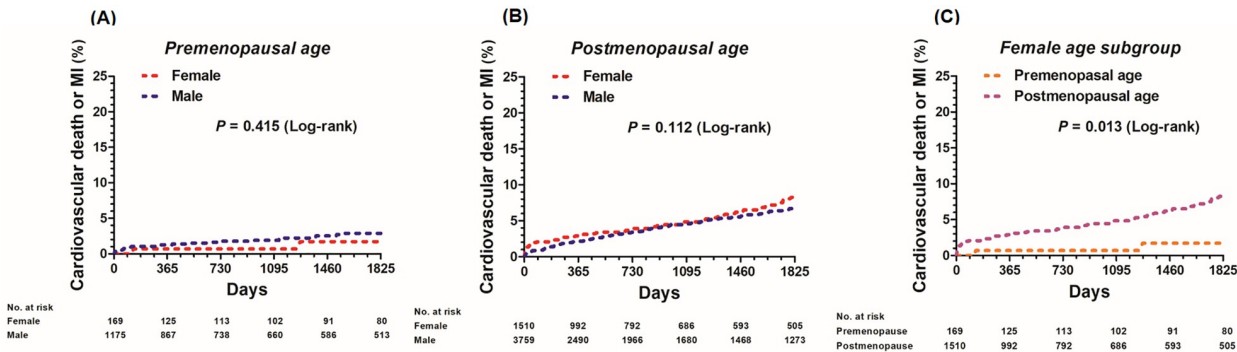

**Fig 3. Kaplan-Meier curves comparing 5-year risk of cardiovascular death or MI in female patients undergoing CABG by age.**
CABG = coronary artery bypass grafting, MI = myocardial infarction.

years between female and male patients treated with CABG, despite female sex being associated with a significantly higher risk of in-hospital and short-term mortality. Associations between sex and mid-term prognosis were maintained after propensity score-matched analysis and were consistent across subgroups by age as well as various clinical factors.

The effect of a patient's sex on prognosis after CABG remains controversial. Several studies reported that various sex-specific factors could affect post-operative outcomes. Blasberg et al. showed that because women had a smaller body surface area and smaller epicardial coronary arteries compared with men, technical difficulties during the CABG procedure lead to higher post-operative risk for poor patency of the surgical graft in women [6]. Furthermore, women more often have microvascular dysfunction, vasospasm, spontaneous coronary artery

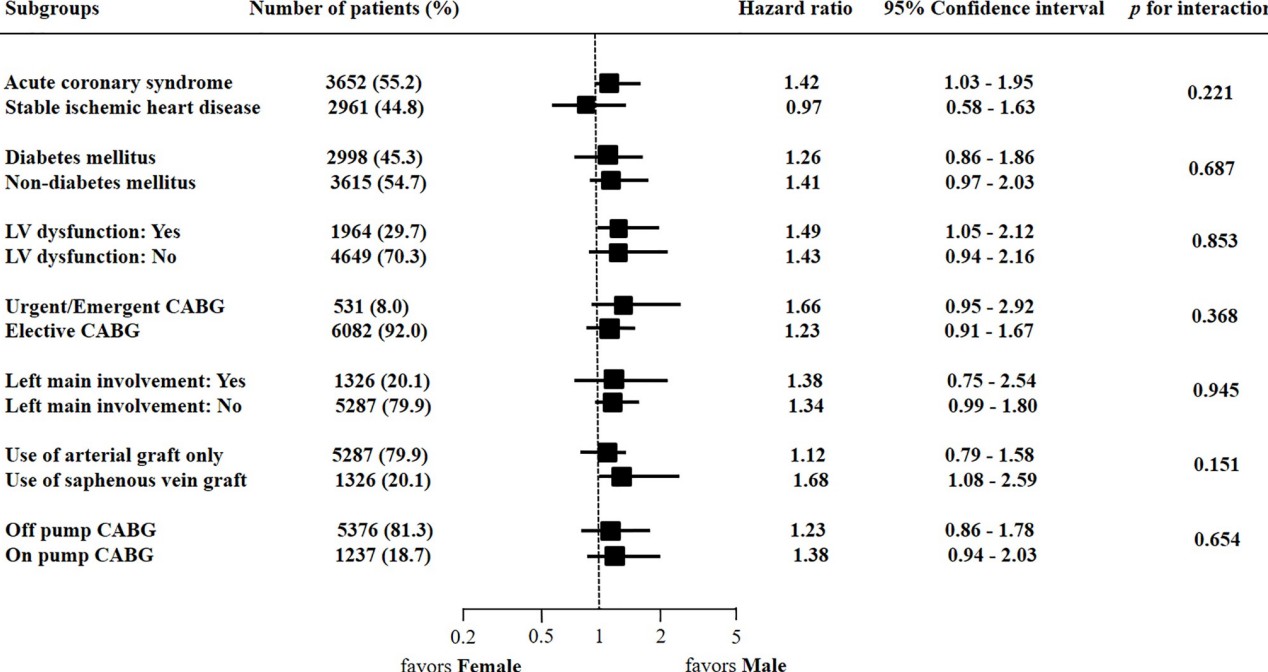

| Subgroups | Number of patients (%) | Hazard ratio | 95% Confidence interval | p for interaction |
|---|---|---|---|---|
| Acute coronary syndrome | 3652 (55.2) | 1.42 | 1.03 - 1.95 | 0.221 |
| Stable ischemic heart disease | 2961 (44.8) | 0.97 | 0.58 - 1.63 | |
| Diabetes mellitus | 2998 (45.3) | 1.26 | 0.86 - 1.86 | 0.687 |
| Non-diabetes mellitus | 3615 (54.7) | 1.41 | 0.97 - 2.03 | |
| LV dysfunction: Yes | 1964 (29.7) | 1.49 | 1.05 - 2.12 | 0.853 |
| LV dysfunction: No | 4649 (70.3) | 1.43 | 0.94 - 2.16 | |
| Urgent/Emergent CABG | 531 (8.0) | 1.66 | 0.95 - 2.92 | 0.368 |
| Elective CABG | 6082 (92.0) | 1.23 | 0.91 - 1.67 | |
| Left main involvement: Yes | 1326 (20.1) | 1.38 | 0.75 - 2.54 | 0.945 |
| Left main involvement: No | 5287 (79.9) | 1.34 | 0.99 - 1.80 | |
| Use of arterial graft only | 5287 (79.9) | 1.12 | 0.79 - 1.58 | 0.151 |
| Use of saphenous vein graft | 1326 (20.1) | 1.68 | 1.08 - 2.59 | |
| Off pump CABG | 5376 (81.3) | 1.23 | 0.86 - 1.78 | 0.654 |
| On pump CABG | 1237 (18.7) | 1.38 | 0.94 - 2.03 | |

favors **Female**          favors **Male**

**Fig 4. Comparative unadjusted hazard ratios of cardiovascular death or MI at 5 years for various subgroups.** CABG = coronary artery bypass grafting, LV = left ventricular.

dissection, or a myocardial bridge than men, and this also could increase perioperative risk [17, 18]. Some studies reported that women are more likely to be referred later in the course of disease and undergo urgent or emergent CABG compared with men. These studies suggested that several clinical factors such as coronary vessel size, body surface area, or atypical symptoms are associated with the higher percentage of emergent procedures or delayed treatment in women, which may explain the higher postoperative and 30-day mortalities in women [5, 7]. Advances in surgical techniques and diagnostic modalities have overcome these clinical issues and many recent studies have shown that smaller vessel size and challenges associated with diagnosis do not contribute to differences in clinical outcomes after CABG [10–12]. Koch et al. showed that females were at a higher risk for post-operative morbidity and in-hospital mortality after CABG, but they evaluated only 25% of women in the study population based on propensity scores that were matched with men [19]. Saxena et al. reported that women who underwent CABG had more comorbidities compared with men, and this was associated with worse outcomes and a higher risk for post-operative complications and in-hospital mortality [4]. However, these reports were based on past data that differed from current practices, and did not identify any differences in mortality on multiple regression analysis. Gaudino et al. recently reported that women have worse outcomes than men in the first 5 years after CABG using data derived from patient-level meta-analysis of four randomized controlled trials [8]. However, in this landmark analysis, the difference in outcomes between sex groups was clearly attenuated after excluding the early post-operative period. Similarly, we found that in-hospital and 30-day mortality after CABG differed between the female and male groups, but these differences in early mortality gradually decreased during follow-up and sex disparity was not associated with clinical outcomes at 5 years. Compared to previous meta-analysis data, a major strength of the current study was the comprehensive evaluation of procedural details, which facilitated adjustment for baseline differences using a vigorous statistical method. Improvements in surgical procedures, advances in the quality of postoperative care, early detection of coronary artery disease, and optimal combinations of CABG and medication (e.g., antiplatelet/anticoagulant, beta-blocker, angiotensin converting enzyme inhibitor or angiotensin receptor blocker, and statin) might have attenuated prognostic differences between males and females [15].

A recent patient-level meta-analysis revealed significant outcome differences by sex between younger patients (<75 years), but not in older patients [8]. However, in our study, sex differences did not affect mid-term clinical outcomes after CABG regardless of menopausal age. Among females, post-menopausal aged patients had a higher incidence of cardiovascular death or MI at 5 years compared to pre-menopausal aged patients. Women generally live longer than men and comorbidities in elderly women are greater in number; however, in comparison, young and middle-aged women are generally healthier than men of the same age. Overall life expectancy has increased consistently due to medical and scientific developments, which will likely result in many elderly male patients that have comorbidities similar to elderly women [14, 20]. Female sex may not be a risk factor for a poor prognosis after CABG and advanced age itself affects many comorbidities and an individual's prognosis regardless of sex. Reports have noted worse clinical profiles among women undergoing CABG with fewer evidence-based treatments provided, which may explain the prognostic role of sex observed in previous studies. Targeted quality improvement treatments may be warranted to narrow sex-related disparities in the quality of care and outcomes among the patients [21, 22].

## Study limitation

There were some limitations to this study. First, this was a non-randomized, retrospective, and observational study, and thus confounding factors or selection bias may have significantly

affected the results. In particular, the choice of surgical technique and the concomitant use of perioperative medications were at the discretion of the individual operator. Although we performed sensitivity analyses, including multivariable Cox regression and propensity score matching, to reduce the effects of potential confounders, we could not adjust for unmeasured variables. Moreover, this study included patients from a single center only. Despite these limitations, the current study analyzed mid-term follow-up data included a large population, and strictly adjusted for confounding factors using propensity score matching analysis. Second, because of the retrospective nature of our registry, we could not thoroughly identify any surgical risk stratification score and alterations in surgical procedure, post-operative treatment or medical therapy in all of the study patients during follow-up. Moreover, we also did not have any data on factors that could induce menopause such as genetics, immune system disorders, medical procedures, and premature ovarian failure and any information about socioeconomic variables, reproductive history, or behavioral and psychosocial characteristics. Therefore, due to the lack of such data, we were unable to determine whether these factors could play a role in the prognostic differences that were observed. Third, our study population had a high prevalence of unstable angina compared with other CABG registries [13, 15], and the findings may not be generalizable to populations with less severe disease. Finally, the high rates of complete arterial graft and off-pump surgery seen in our hospital and the low rates of peri-procedural MI may have been affected by racial differences or underestimation, and this may further limit the generalizability of our results.

## Conclusion

In patients treated with CABG, early post-operative complications were more frequent in women than in men; however, there was no significant difference in the mid-term risk of cardiovascular death or MI and secondary outcomes between female and male groups, regardless of menopausal age. Based on our results, sex does not seem to influence mid-term clinical outcomes in patients who undergo CABG. Further investigation regarding the potential therapeutic implications of these findings to narrow sex-related prognosis and disparities should be considered.

## Supporting information

**S1 Table. Clinical outcomes at 5 years in patients treated with isolated CABG.**
(DOC)

**S1 Fig. The incidences of cardiac death or myocardial infarction (MI) by era of coronary artery bypass grafting surgery.**
(TIFF)

**S1 Data.**
(XLS)

## Author Contributions

**Conceptualization:** Woo Jin Jang, Ki Hong Choi, Jeong Hoon Yang, Joo-Yong Hahn, Seung-Hyuk Choi, Hyeon-Cheol Gwon, Yang Hyun Cho, Kiick Sung, Wook Sung Kim, Dong Seop Jeong, Young Bin Song.

**Data curation:** Woo Jin Jang, Ki Hong Choi, Jihoon Kim, Jeong Hoon Yang, Joo-Yong Hahn, Seung-Hyuk Choi, Hyeon-Cheol Gwon, Yang Hyun Cho, Kiick Sung, Wook Sung Kim, Dong Seop Jeong, Young Bin Song.

**Formal analysis:** Woo Jin Jang, Joo-Yong Hahn, Seung-Hyuk Choi, Hyeon-Cheol Gwon, Yang Hyun Cho, Kiick Sung, Wook Sung Kim, Dong Seop Jeong, Young Bin Song.

**Investigation:** Woo Jin Jang, Ki Hong Choi, Jihoon Kim, Jeong Hoon Yang, Joo-Yong Hahn, Seung-Hyuk Choi, Hyeon-Cheol Gwon, Yang Hyun Cho, Kiick Sung, Wook Sung Kim, Dong Seop Jeong, Young Bin Song.

**Methodology:** Woo Jin Jang, Ki Hong Choi, Jihoon Kim, Young Bin Song.

**Resources:** Jihoon Kim, Young Bin Song.

**Supervision:** Woo Jin Jang, Young Bin Song.

**Validation:** Woo Jin Jang, Ki Hong Choi, Young Bin Song.

**Visualization:** Woo Jin Jang.

**Writing – original draft:** Woo Jin Jang, Ki Hong Choi.

**Writing – review & editing:** Woo Jin Jang, Young Bin Song.

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
