## [Decision Letter · Decision Letter 0]

25 May 2022

PONE-D-22-08451Prognostic role of Sex in Coronary Artery Bypass GraftingPLOS ONE

Dear Dr. Song,

Thank you for submitting your manuscript to PLOS ONE. After careful consideration, we feel that it has merit but does not fully meet PLOS ONE’s publication criteria as it currently stands. Therefore, we invite you to submit a revised version of the manuscript that addresses the points raised during the review process.

ACADEMIC EDITOR: 

Authors are encouraged to follow the suggestions made by Reviewer #1

We look forward to receiving your revised manuscript.

Kind regards,

Antonino Salvatore Rubino, M.D., Ph.D.

Academic Editor

PLOS ONE

Journal Requirements:

Additional Editor Comments:

Authors are encouraged to follow the suggestions made by Reviewer#1

Reviewers' comments:

Reviewer's Responses to Questions

**Comments to the Author**

1. Is the manuscript technically sound, and do the data support the conclusions?

Reviewer #1: Partly

2. Has the statistical analysis been performed appropriately and rigorously? 

Reviewer #1: I Don't Know

3. Have the authors made all data underlying the findings in their manuscript fully available?

Reviewer #1: Yes

4. Is the manuscript presented in an intelligible fashion and written in standard English?

Reviewer #1: Yes

5. Review Comments to the Author

Reviewer #1: Have done well to use Propensity Score Matching (PSM) as a tool to study the influence of Sex/Gender on the short and longer term outcome following CABG. A number of PSM studies agree with your finds while others refute it. More recently Chang F-C, et al. BMJ Open 2022;12:e058538 suggests that Female Sex still has a significantly negative impact on the outcome of CABG as well as other major cardiac surgery in a larger cohort of patients (more than 60,000 cases).

Also review ... Alam et al. Am J Cardiology. 2013 Aug 1;112(3):309-17

…. women who underwent isolated CABG experienced higher mortality at short-term, midterm, and long-term follow-up compared with men. Mortality remained independently associated with female gender despite propensity score-matched analysis of outcomes….

and

Guadino et al EHJ. 2021 Dec 28;43(1):18-28.

6. PLOS authors have the option to publish the peer review history of their article (what does this mean?). If published, this will include your full peer review and any attached files.

Reviewer #1: **Yes: **Prof. Reida M El Oakley FRCS, MD - https://orcid.org/0000-0003-4101-8998

---

## [Author Response · Author response to Decision Letter 0]

2 Jul 2022

Response to Reviewer #1 Comments

First of all, I behalf of all authors thank the reviewer #1 for positive comments.

Response Table. Comparisons of All-cause mortality after isolated CABG in studies 

Study Study duration Type of Analysis Clinical outcome Total population Women (n/N, %) Men (n/N, %) adjusted HR (95% CI) p value

Jang et al. (PONE-D-22-08451, The present study) 2001 ~ 2017 PS-matched analysis 1-year cardiac mortality 6613 66/1679, 6.1% 134/4934, 4.4% 1.14 (0.78 - 1.67) 0.507 

 1-year all-cause mortality 116/1679, 10.9% 272/4934, 8.7% 1.02 (0.77 - 1.34) 0.904 

Alam et al. Am J Cardiology 2013 Aug 1; 112(3):309-17 1982 ~ 2012 PS- matched meta-analysis 1-year all-cause mortality 92660 504/19780, 2.6% 1420/72880, 1.9% 1.31 (1.18 - 1.45) 0.710 

Guadino et al. EHJ. 2021 Dec 28;43(1):18-28 2001 ~ 2012 Multivariable adjusted 

Cox meta-analysis 5-year all-cause mortality 13193 558/2714, 20.6% 1817/10479, 17.3% 1.03 (0.94 - 1.14) 0.510 

Chang F-C et al. BMJ Open 2022;12:e058538 2000 ~ 2013 PS-matched analysis Long-term all-cause mortality 42125 4380/18720, 44.6% 4453/18720, 45.4% 0.98 (0.95 - 1.02) 0.429 

CABG = coronary artery bypass grafting surgery, CI = confidence interval, HR = hazard ratio, PS = propensity score.

As the reviewer #1’s comment, the prognostic effect of sex in patients treated with isolated coronary artery bypass grafting (CABG) still is controversy. Chang F-C at al. evaluated the sex-related differences in clinical outcomes after major adult cardiac surgery using nationwide, population-based cohort study [1]. They reported that female patients who underwent CABG (with or without concurrent valvular intervention) generally had worse in-hospital complication (composite of in-hospital mortality, new-onset stroke and new onset dialysis during the admission for the index surgery) and late outcomes (MACE; composite of cardiovascular death, revascularization, acute myocardial infraction or ischaemic stroke during follow-up) but in the isolated CABG patients, they showed no difference in long-term all-cause mortality between women and men same as the results of our study (Response Table). Alam et al. also reported that women who underwent isolated CABG experienced higher mortality at short-term, midterm, and long-term follow-up compared with men but most of their data are very old, including even data before 1990 [2], therefore their conclusion cannot reflect current clinical practice. Especially, because of the incidence of very low incidences of all clinical outcomes (Response Table), the large differences in baseline characteristics between the genders, and suboptimal risk adjustment due to low-quality data, the Alam’s study has considerable limitations. Guadino et al. reported worse MACCE (composite of all-cause mortality, myocardial infarction, stroke, and repeat revascularization) in women over 5 years of follow-up compared to men [3] however they also showed similar all-cause mortality between men and women (adjusted HR 1.03, 95% CI 0.94–1.14; p = 0.510) (Response Table). Despite of heterogeneity between several studies about gender disparity, the similarity in long-term mortality is consistent between female and male patients undergoing isolated CABG as in our study. Based on our results and former studies, sex does not seem to influence long-term clinical outcomes, especially long-term mortality in patients who undergo CABG. Further understanding the gender differences in demographics and surgical characteristics can narrow the gap of clinical outcomes and improve their prognosis.

To respond the reviewer #1’s comment, we revised references about prognostic role of gender disparity in CABG and added following new references in the revised manuscript.

(INTRODUCTION, page 4, line 8 and REFERENCES, page 13, line 20-25 in the revised manuscript)

8. Gaudino M, Di Franco A, Alexander JH, Bakaeen F, Egorova N, Kurlansky P, et al. Sex differences in outcomes after coronary artery bypass grafting: a pooled analysis of individual patient data. Eur Heart J. 2021;43(1):18-28. Epub 2021/08/03. doi: 10.1093/eurheartj/ehab504. PubMed PMID: 34338767; PubMed Central PMCID: PMCPMC8851663.

9. Chang FC, Chen SW, Chan YH, Lin CP, Wu VC, Cheng YT, et al. Sex differences in risks of in-hospital and late outcomes after cardiac surgery: a nationwide population-based cohort study. BMJ Open. 2022;12(2):e058538. Epub 2022/02/04. doi: 10.1136/bmjopen-2021-058538. PubMed PMID: 35110325; PubMed Central PMCID: PMCPMC8811586.

We would like to sincerely thank reviewer #1 for valuable comments. Addressing them fully has significantly strengthened the manuscript.

REFERENCES

1. Chang FC, Chen SW, Chan YH, Lin CP, Wu VC, Cheng YT, et al. Sex differences in risks of in-hospital and late outcomes after cardiac surgery: a nationwide population-based cohort study. BMJ Open. 2022; 12:e058538. https://doi.org/10.1136/bmjopen-2021-058538 PMID: 35110325

2. Alam M, Bandeali SJ, Kayani WT, Ahmad W, Shahzad SA, Jneid H, et al. Comparison by meta-analysis of mortality after isolated coronary artery bypass grafting in women versus men. Am J Cardiol. 2013; 112:309-17. https://doi.org/10.1016/j.amjcard.2013.03.034 PMID: 23642381

3. Gaudino M, Di Franco A, Alexander JH, Bakaeen F, Egorova N, Kurlansky P, et al. Sex differences in outcomes after coronary artery bypass grafting: a pooled analysis of individual patient data. Eur Heart J. 2021; 43:18-28. https://doi.org/10.1093/eurheartj/ehab504 PMID: 34338767

---

## [Decision Letter · Decision Letter 1]

20 Oct 2022

PONE-D-22-08451R1

Prognostic role of Sex in Coronary Artery Bypass Grafting

PLOS ONE

Dear Dr. Song,

Thank you for submitting your manuscript to PLOS ONE. After careful consideration, we feel that it has merit but does not fully meet PLOS ONE’s publication criteria as it currently stands. Therefore, we invite you to submit a revised version of the manuscript that addresses the points raised during the review process.

We look forward to receiving your revised manuscript.

Kind regards,

Shukri AlSaif

Academic Editor

PLOS ONE

Reviewers' comments:

Reviewer's Responses to Questions

**Comments to the Author**

1. If the authors have adequately addressed your comments raised in a previous round of review and you feel that this manuscript is now acceptable for publication, you may indicate that here to bypass the “Comments to the Author” section, enter your conflict of interest statement in the “Confidential to Editor” section, and submit your "Accept" recommendation.

Reviewer #1: (No Response)

Reviewer #2: (No Response)

2. Is the manuscript technically sound, and do the data support the conclusions?

Reviewer #1: Yes

Reviewer #2: Yes

3. Has the statistical analysis been performed appropriately and rigorously? 

Reviewer #1: I Don't Know

Reviewer #2: Yes

4. Have the authors made all data underlying the findings in their manuscript fully available?

Reviewer #1: Yes

Reviewer #2: Yes

5. Is the manuscript presented in an intelligible fashion and written in standard English?

Reviewer #1: Yes

Reviewer #2: Yes

6. Review Comments to the Author

Reviewer #1: Thank you for the changes you made.

Reviewer #2: The authors attempt to address the controversial issue of impact of gender on long-term prognosis of patients who underwent coronary artery bypass grafting (CABG). They evaluated their institutional database and concluded that after adjusting for baseline differences, sex does not appear to influence long-term risk of cardiovascular death or myocardial infarction (MI) in patients undergoing CABG. In the opinion of this reviewer the quality of the manuscript will be further enhanced if the authors address the following issues.

[1] The title needs to be revised. It will be better to rephrase the title as: Impact of gender on mid-term prognosis of patients undergoing isolated coronary artery bypass grafting.

[2] Abstract, Objective: preferably 2 to 3 sentences about the targeted patient population, the current management or treatment controversies and the primary objective(s) of the study.

[3] Abstract, Methods: Please mention study design.

[4] Please exclude patients with combined procedures and only include a pure cohort of patients who underwent isolated CABG.

[5] What risk stratification score (EuroSCORE, STS or Parsonnet) was used for this cohort? Please provide this information.

[6] The data was censored at 5 years. Generally 5 years is regarded as mid-term outcome. It is important to replace long-term with mid-term in the title and throughout the manuscript to reflect this fact.

[7] Was there any missing data? If so how was it addressed?

[8] The study period spans 16 years. It is essential to investigate the effect of era of surgery on outcomes.

[9] Please add a few sentences about the unanswered questions and future research.

7. PLOS authors have the option to publish the peer review history of their article (what does this mean?). If published, this will include your full peer review and any attached files.

Reviewer #1: **Yes: **Professor Reida El Oakley FRCS, MD

Reviewer #2: **Yes: **Shahzad G. Raja

---

## [Author Response · Author response to Decision Letter 1]

21 Nov 2022

- RESPONSE TO REVIEWERS -

Reviewer #1: Thank you for the changes you made.

Response to Reviewer #1 Comments

Dear Professor Reida El Oakley FRCS, MD

I behalf of all co-authors sincerely thank you for your positive comments about our work and also thank you again for the constructive and detailed previous review.

Reviewer #2: The authors attempt to address the controversial issue of impact of gender on long-term prognosis of patients who underwent coronary artery bypass grafting (CABG). They evaluated their institutional database and concluded that after adjusting for baseline differences, sex does not appear to influence long-term risk of cardiovascular death or myocardial infarction (MI) in patients undergoing CABG. In the opinion of this reviewer the quality of the manuscript will be further enhanced if the authors address the following issues.

[1] The title needs to be revised. It will be better to rephrase the title as: Impact of gender on mid-term prognosis of patients undergoing isolated coronary artery bypass grafting.

[2] Abstract, Objective: preferably 2 to 3 sentences about the targeted patient population, the current management or treatment controversies and the primary objective(s) of the study.

[3] Abstract, Methods: Please mention study design.

[4] Please exclude patients with combined procedures and only include a pure cohort of patients who underwent isolated CABG.

[5] What risk stratification score (EuroSCORE, STS or Parsonnet) was used for this cohort? Please provide this information.

[6] The data was censored at 5 years. Generally 5 years is regarded as mid-term outcome. It is important to replace long-term with mid-term in the title and throughout the manuscript to reflect this fact.

[7] Was there any missing data? If so how was it addressed?

[8] The study period spans 16 years. It is essential to investigate the effect of era of surgery on outcomes.

[9] Please add a few sentences about the unanswered questions and future research.

Response to Reviewer #2 (Dr. Shahzad G. Raja)’s Comments

[1] The title needs to be revised. It will be better to rephrase the title as: Impact of gender on mid-term prognosis of patients undergoing isolated coronary artery bypass grafting.

→Response #1

As the Reviewer #2’s comment, we modified our full title and short title in the revised manuscript as follows (Title page, page 1, line 1-3 in the revised manuscript).

Prognostic role of Sex in Coronary Artery Bypass Grafting

Short title: Sex and Coronary artery bypass grafting

→ Impact of Gender on mid-term prognosis of patients undergoing Coronary Artery Bypass Grafting

Short title: Gender and Coronary artery bypass grafting

[2] Abstract, Objective: preferably 2 to 3 sentences about the targeted patient population, the current management or treatment controversies and the primary objective (s) of the study.

→Response #2

As the Reviewer #2’s comment, we added following sentences in the revised manuscript (ABSTRACT, Objectives, page 4, line 3-4 in the revised manuscript).

Data on gender differences in current management or clinical outcomes after CABG are controversial, and there have been limited data focusing on them.

[3] Abstract, Methods: Please mention study design.

→Response #3

As the Reviewer #2’s comment, we added following sentences about the study design in the revised manuscript (ABSTRACT, Methods, page 4, line 5-7 in the revised manuscript).

Methods: Between January 2001 and December 2017, 6613 patients who underwent CABG were enrolled and divided into two groups according to sex (female group, n = 1,679 vs. male group, n = 4,934).

→ Methods: This was a retrospective, single-center, observational study. Between January 2001 and December 2017, 6613 consecutive patients who underwent CABG were enrolled from an institutional registry of Samsung Medical Center, Seoul, Korea (Clinicaltrials.gov, NCT03870815) and divided into two groups according to sex (female group, n = 1679 vs. male group, n = 4934).

[4] Please exclude patients with combined procedures and only include a pure cohort of patients who underwent isolated CABG.

→Response #4

I agree with Reviewer #2’s concern about clinical effect of combined other procedures in patients treated with CABG. However, as we already mentioned in the original manuscript, our study population included patients with coronary artery disease who underwent CABG including only 424 patients (6.4% of total population) treated with combined valvular surgery (Table 2). As the reviewer’s comment, we additionally evaluated data excluding these 424 patients but clinical outcomes in 6189 isolated CABG patients showed as follows.

Supplemental table 1. Clinical outcomes at 5 years in patients treated with isolated CABG

　 Female Male Univariable analysis P-value ¹Multivariable analysis P-value

 HR 95% CI HR 95% CI 

 n = 1544 n = 4645 　 　

Cardiovascular death or myocardial infarction 62 (4.0) 155 (3.3) 1.22 0.91 - 1.64 0.185 0.99 0.72 - 1.37 0.953 

Cardiovascular death 51 (3.3) 121 (2.6) 1.29 0.93 - 1.78 0.133 0.96 0.67 - 1.37 0.831 

All-cause death 90 (5.8) 242 (5.2) 1.14 0.89 - 1.45 0.302 0.86 0.67 - 1.12 0.272 

Myocardial infarction 15 (1.0) 42 (0.9) 1.09 0.61 - 1.97 0.772 1.14 0.59 - 2.20 0.695 

Stroke 31 (2.0) 103 (2.2) 0.91 0.61 - 1.36 0.657 0.84 0.55 - 1.31 0.448 

BARC type 3-5 bleeding 38 (2.5) 115 (2.5) 1.05 0.74 - 1.50 0.796 0.9 0.61 - 1.32 0.578 

²MACE 94 (6.1) 260 (5.6) 1.05 0.86 - 1.29 0.610 0.87 0.70 - 1.09 0.874 

Values are n (%). Cumulative incidence of events was presented as Kaplan–Meier estimates.

ACE = angiotensin converting enzyme, ARB = angiotensin receptor blocker, BARC = bleeding academic research consortium, CABG = coronary artery bypass grafting surgery, CI = confidence interval, HR = hazard ratio, LITA = left internal thoracic artery, MACE = major adverse cardiovascular event, MI = myocardial infarction, RITA = right internal thoracic artery, SVG = saphenous vein graft.

¹Adjusted variables included age, body mass index, hypertension, diabetes mellitus, current smoking, heart failure, previous history of MI, use of antiplatelet, use of beta-blocker, use of ACE inhibitor or ARB, use of statin, multi-vessel disease, left main involvement, off-pump CABG, combined valvular surgery, number of anastomosis, use of LITA, use of RITA, use of bilateral thoracic arteries, and use of SVG.

²MACE was defined as the composite of cardiovascular death, myocardial infarction, and stroke.

There are similar between the results of original study population and isolated CABG subgroup. Current treatment of CABG performed in accordance with relevant standard guidelines [1] frequently involved combined valvular disease. Therefore, we have thought that data of CABG including combined valvular surgery reflect real-world practice better and it is more meaningful to potential your reader.

To respond the Reviewer #2’s comment, we added above table as supplemental table in the revised manuscript (SUPPLEMENTAL MATERIAL, Supplemental table1, page 2 line 1- page 3, line 7 in the revised manuscript).

REFERENCE

1. Neumann FJ, Sousa-Uva M, Ahlsson A, Alfonso F, Banning AP, Benedetto U, et al. 2018 ESC/EACTS Guidelines on myocardial revascularization. Eur Heart J. 2019; 40:87-165. https://doi.org/10.1093/eurheartj/ehy394 PMID: 30165437

[5] What risk stratification score (EuroSCORE, STS or Parsonnet) was used for this cohort? Please provide this information.

→Response #5

This is the important point of view. Unfortunately, due to the retrospective nature of our registry, we do not have detailed data regarding the surgical risk stratification score such as EuroSCORE, STS or Parsonnet. To the Reviewer #2’s comment, we added following comment in our study limitations (DISCUSSION, Study limitations, page 24, line 9 in the revised manuscript).

Second, because of the retrospective nature of our registry, we could not thoroughly identify alterations in post-operative treatment or medical therapy in all of the study patients during follow-up. Moreover, we also did not have any data on factors that could induce menopause such as genetics, immune system disorders, medical procedures, and premature ovarian failure and any information about socioeconomic variables, reproductive history, or behavioral and psychosocial characteristics. Therefore, due to the lack of such data, we were unable to determine whether these factors could play a role in the prognostic differences that were observed.

→Second, because of the retrospective nature of our registry, we could not thoroughly identify any surgical risk stratification score and alterations in surgical procedure, post-operative treatment, or medical therapy in all of the study patients during follow-up. Moreover, we also did not have any data on factors that could induce menopause such as genetics, immune system disorders, medical procedures, and premature ovarian failure and any information about socioeconomic variables, reproductive history, or behavioral and psychosocial characteristics. Therefore, due to the lack of such data, we were unable to determine whether these factors could play a role in the prognostic differences that were observed.

[6] The data was censored at 5 years. Generally 5 years is regarded as mid-term outcome. It is important to replace long-term with mid-term in the title and throughout the manuscript to reflect this fact.

→Response #6

As the Reviewer #2’s comment, we revised “long-term” to “mid-term” in the title, abstract, and text throughout the whole manuscript as follows. 

long-term →mid-term

(ABSTRACT, Objectives, page 4, line 2 in the revised manuscript), (INTRODUCTION, page 7, line 12 in the revised manuscript), (DISCUSSION, page 21, line 6, line 7, and line 9 in the revised manuscript), (DISCUSSION, page 23, line 8 in the revised manuscript), (DISCUSSION, study limitation, page 24, line 8 in the revised manuscript), and (CONCLUSION, page 25, line 5 in the revised manuscript)

[7] Was there any missing data? If so how was it addressed?

→Response #7

Thank you for your valuable comment. As you know, since this study was conducted from single-center data, missing data could be minimized through a comprehensive chart review. Therefore, no imputation methods were used to infer missing values of baseline variables.

[8] The study period spans 16 years. It is essential to investigate the effect of era of surgery on outcomes.

→Response #8

This is also the excellent point of view. As the Reviewer #2’ comment, we additionally investigated the gender difference of primary outcome (cardiac death or MI) by era of CABG in our study population.

The incidences of cardiac death or MI after CABG had improved for both female and male group over time but irrespective of the era of CABG, there are no significant differences in the incidence of cardiac death or MI between female and male groups. To respond the Reviewer #2’s comment, we added above figure as supplemental figure (SUPPLEMENTAL MATERIAL, Supplemental figure 1, page 4 line 1-3 in the revised manuscript) and revised sentences in the study limitations of revised manuscript as follows (DISCUSSION, Study limitations, page 24, line 10 in the revised manuscript).

Second, because of the retrospective nature of our registry, we could not thoroughly identify alterations in post-operative treatment or medical therapy in all of the study patients during follow-up. Moreover, we also did not have any data on factors that could induce menopause such as genetics, immune system disorders, medical procedures, and premature ovarian failure and any information about socioeconomic variables, reproductive history, or behavioral and psychosocial characteristics. Therefore, due to the lack of such data, we were unable to determine whether these factors could play a role in the prognostic differences that were observed.

→Second, because of the retrospective nature of our registry, we could not thoroughly identify any data about surgical risk and alterations in surgical procedure, post-operative treatment, or medical therapy in all of the study patients during follow-up. Moreover, we also did not have any data on factors that could induce menopause such as genetics, immune system disorders, medical procedures, and premature ovarian failure and any information about socioeconomic variables, reproductive history, or behavioral and psychosocial characteristics. Therefore, due to the lack of such data, we were unable to determine whether these factors could play a role in the prognostic differences that were observed.

[9] Please add a few sentences about the unanswered questions and future research.

→Response #9

As the Reviewer #2’s comment, we revised following sentence for more clear mention about the unanswered question and future research in the revised manuscript (CONCULUSION, page 25, line 8-9 in the revised manuscript).

Further investigation regarding the potential therapeutic implications of these findings should be considered.

→ Further investigation regarding the potential therapeutic implications of these findings to narrow sex-related prognosis and disparities should be considered.

We would like to sincerely thank Dr. Shahzad G. Raja for valuable comments. Addressing them fully has significantly strengthened the manuscript.

---

## [Decision Letter · Decision Letter 2]

29 Nov 2022

Impact of Gender on mid-term prognosis of patients undergoing Coronary Artery Bypass Grafting

PONE-D-22-08451R2

Dear Dr. Song,

We’re pleased to inform you that your manuscript has been judged scientifically suitable for publication and will be formally accepted for publication once it meets all outstanding technical requirements.

Please address the comments raised by Reviewer 3 by doing the appropriate corrections.

Thank you.

Kind regards,

Shukri AlSaif

Academic Editor

PLOS ONE

Additional Editor Comments (optional):

Reviewers' comments:

Reviewer's Responses to Questions

**Comments to the Author**

1. If the authors have adequately addressed your comments raised in a previous round of review and you feel that this manuscript is now acceptable for publication, you may indicate that here to bypass the “Comments to the Author” section, enter your conflict of interest statement in the “Confidential to Editor” section, and submit your "Accept" recommendation.

Reviewer #2: All comments have been addressed

Reviewer #3: All comments have been addressed

2. Is the manuscript technically sound, and do the data support the conclusions?

Reviewer #2: Yes

Reviewer #3: Yes

3. Has the statistical analysis been performed appropriately and rigorously? 

Reviewer #2: Yes

Reviewer #3: Yes

4. Have the authors made all data underlying the findings in their manuscript fully available?

Reviewer #2: Yes

Reviewer #3: Yes

5. Is the manuscript presented in an intelligible fashion and written in standard English?

Reviewer #2: Yes

Reviewer #3: Yes

6. Review Comments to the Author

Reviewer #2: Thank you for addressing all the highlighted issues. The manuscript will add further to the existing literature on the subject.

Reviewer #3: I am reviewing a revision of this paper having not reviewed the original submission, but note the comments of the previous reviewers which the authors have responded to.

They present a propensity matched analysis of a retrospectively collected registry from a single centre over 17 years to assess the mid-term outcomes of male vs female patients. The statistical analysis appears appropriate and the paper is written to a very high standard of English. The comments of the previous reviewers did not require significant changes as this appears to have been a meticulously constructed paper from the outset, but the responses provided are all appropriate and complete.

A few minor comments.

1) The registry entry on clinicaltrials.gov is made to sound like the registration for this study. I suggest rewording to emphasise that this is a subset of a previously acquired dataset from a comparison of CABG and PCI.

2) Similarly, the suggestion that data was collected prospectively by research coordinators also needs to be revised - the NCT shows this was retrospective data

Otherwise, this is a well written paper and will contribute valuably to the literature.

7. PLOS authors have the option to publish the peer review history of their article (what does this mean?). If published, this will include your full peer review and any attached files.

Reviewer #2: **Yes: **Shahzad G. Raja

Reviewer #3: No

---

## [Editor Report · Acceptance letter]

22 Feb 2023

PONE-D-22-08451R2 

Impact of Gender on mid-term prognosis of patients undergoing Coronary Artery Bypass Grafting 

Dear Dr. Song:

I'm pleased to inform you that your manuscript has been deemed suitable for publication in PLOS ONE. Congratulations! Your manuscript is now with our production department. 

Kind regards, 

on behalf of

Dr. Shukri AlSaif 

Academic Editor

PLOS ONE